# Temperate Grassland Afforestation Dynamics in the Aguapey Valuable Grassland Area between 1999 and 2020: Identifying the Need for Protection

**Melisa Apellaniz \*, Niall G. Burnside**  **and Matthew Brolly**

Centre for Earth Observation Science, University of Brighton, Brighton BN2 4AT, UK;
N.G.Burnside@brighton.ac.uk (N.G.B.); M.Brolly@brighton.ac.uk (M.B.)
**\*** Correspondence: melapel@gmail.com

**Abstract:** Temperate grasslands are considered the most endangered terrestrial ecosystem worldwide; the existent areas play a key role in biodiversity conservation. The Aguapey Valuable Grassland Area (VGA), one of the most well-preserved temperate grassland areas within Argentina, is currently threatened by the anthropogenic expansion of exotic tree plantations. Little is known about the impacts of afforestation over temperate grassland landscape structures; therefore, the aim of this study is to characterize Aguapey VGA landscape structural changes between 1999 and 2020 based on remotely sensed data. This involves the generation of land cover maps for four annual periods based on unsupervised classification of Landsat 5 TM and 8 OLI images, the estimation of landscape metrics, and the transition analysis between land cover types and annual periods. The area covered by temperate grassland is shown to have decreased by almost 22% over the 20 year-period studied, due to the expansion of tree plantation cover. The afforestation process took place mainly between 1999 and 2007 in the northern region of the Aguapey VGA, which led first to grassland perforation and subsequently to grassland attrition; however, Aguapey's cultural tradition of cattle ranching could have partially inhibited the expansion of exotic trees over the final years of the study. The evidence of grassland loss and fragmentation within the Aguapey VGA should be considered as an early warning to promote the development of sustainable land use policies, mainly focused towards the Aguapey VGA's southern region where temperate grassland remains the predominant land cover type.

**Keywords:** temperate grassland; afforestation; fragmentation; land cover change; landscape dynamics



## 1. Introduction

Temperate grasslands occupy circa 8% of the Earth's terrestrial surface and are among the most biodiverse and productive ecosystems worldwide [1]. Not only are they a habitat of with high diversity and an abundance of flora and fauna species, including many endemics and endangered species, temperate grasslands are also important for many plant food species of economic importance [2]. Furthermore, they provide many important ecosystems services (e.g., nutrient recycling, pollination, habitat for livestock grazing, genetic diversity for crops, recreation, climate regulation) and play a key role in global carbon cycle: grasslands soils store as much carbon as forests do globally [1,3]. After centuries of human disturbance, temperate grasslands are considered one of the most altered and endangered ecosystems on most continents: about 41% of these grasslands have already been converted to agricultural use, 7.5% to commercial forestry, and almost 6% were replaced by urbanization [1,4]. Despite its critical conservation status, only 4.59% of these grasslands are under some level of protection [5]. In a context of increasing threats by multiple anthropogenic activities, the remaining areas of native temperate grasslands take on a heightened importance for the conservation of biodiversity and ecosystem services that are essential to sustain human development and well-being [3].

The Rio de la Plata Grasslands (RPG) are the main complex of temperate grassland ecosystems in South America, covering the large plains of central-east Argentina, Uruguay, and Southern Brazil. The RPG, originally characterized by the almost absolute absence of trees, are habitat for a conspicuous and unique biodiversity including more than 550 different grass species, 450 bird species, and nearly a hundred species of mammals [6,7]. Significant human transformation across this region started mainly at the beginning of the 20th century, with increasing European immigration and the replacement of native vegetation to agriculture [6]. Over recent decades, the RPG have recorded some of the highest rates of land use change worldwide given their intensified use for livestock production and grasslands' conversion to crops, implanted pastures, and exotic tree plantations [8,9]. Some of the remaining and most well-preserved temperate grassland areas in Argentina are located mostly in private land of the Northern Campos region (the Campos, hereafter) [7,10]. Only 0.15% of Campos grasslands have formal protection designation; therefore, many sites of high biodiversity are in a potentially vulnerable situation [7].

In 1998, Argentina's government enacted a law particularly aimed to promote and financially support the development of forestry plantations in different regions across the country, in order to supply the growing global demand of pulp and wood (Argentina' National Law 25.080). This policy triggered a widespread land-use change across the country that led to the replacement of grassland areas used for cattle ranching activities by exotic tree monocultures [11]. The afforestation development was particularly important across the Campos region given its exceptional climatic and edaphic conditions that prompt high annual growth rates of exotic tree species such as *Pinus* spp. and *Eucalyptus* spp. [11,12].

The severe ecological consequences of the conversion of a grass-dominated ecosystem to one dominated by trees (referred to as grassland afforestation) not only derives from the direct reduction of the original grassland cover [13–15] but also from the indirect transformation of the spatial configuration of the landscape [16,17]. Grassland afforestation typically promotes grassland fragmentation, as the (formerly) locally continuous natural ecosystem is broken into smaller and more isolated fragments surrounded by a human-transformed matrix of tree plantations [1,18,19]. Changes in the size and spatial configuration of remnant fragments are recognized to have a major effect, not only on population dynamics and species persistence [19,20] but also on the ecosystem processes that ultimately determine the provision of ecosystem services [19,21].

Fragmentation is a dynamic process of change that leads to different stages: perforation or incision, dissection, dissipation, shrinkage, and attrition [22,23]. The stage of fragmentation provides critical information not only to infer changes in the ecosystem (i.e., an early fragmentation stage could be interpreted as an early warning sign) but also for the development of suitable ecosystem management strategies [21,23,24]. Despite its importance, studies of landscape fragmentation have typically been biased towards forest ecosystems [24]. Only in recent years have grassland fragmentation studies gained more attention; however, grassland fragmentation induced by afforestation processes remains understudied [25].

Within contemporary climate change mitigation scenarios, the development of the carbon market may accelerate the rate of grassland afforestation in the Campos region [15,26]. Furthermore, as part of Argentina's national contribution to 'the fight against climate change' established in the Paris Agreement (2019), the National Government launched a new initiative to increase tree plantation cover by 50% in the period up to 2030 (which represents a total of 2 million hectares) [27]. This situation has raised particular concern in the Aguapey Valuable Grassland Area (VGA), an area of high biodiversity conservation value within the Campos region [3] which demonstrates favorable conditions for the establishment of tree plantations [13,28].

The potentially severe ecological consequences of the Campos grassland afforestation have started to be increasingly recognized [13,15,29]. However, it is also necessary to further increase the current knowledge about the level of temperate grasslands' fragmentation due to afforestation, in order to achieve a broader understanding of the afforestation

impacts over temperate grasslands and support the development of land-use planning policies with a conservation focus. The development of robust methodologies to determine the current extent, and recent levels of loss and transformation, will be central to the development of successful mitigation measures and to secure the careful custodianship of this internationally important temperate grassland ecosystem in South America.

The aim of this study is to analyze the spatio-temporal changes in the landscape structure of the Aguapey VGA due to the afforestation process occurring between 1999 and 2020; and to understand the nature of landscape transformation across this period. For this purpose, grassland, and tree plantation cover within the Aguapey VGA are characterized in four time periods between 1999 and 2020 (1999–2000; 2006–2007; 2014–2015; 2019–2020) employing an unsupervised classification methodology, based upon a multi-temporal sequence of past and present Landsat multispectral images. Secondly, in order to provide a deeper understanding of the dynamics of change, a land covers' transition probability analysis is made between the temporal periods. Finally, key landscape metrics were estimated to analyze the temporal changes that have occurred to the Aguapey VGA landscape structure, with focus on the loss and fragmentation of temperate grasslands.

## 2. Materials and Methods

### 2.1. Study Area

The Aguapey VGA (1598.11 km$^2$; central coordinates 27°56′S, 56°26″W) is located in the Aguapey basin, within the Campos region of the RPG (Corrientes province, northeast of Argentina) [3] (Figure 1). This area is characterized by a matrix of temperate grasslands mainly dependent on its topographic location. Flat lowlands are dominated by tall-grass 'paja colorada' *Andropogon lateralis*; depressions and drainage areas located towards the Aguapey river's margin are dominated by *Paspalum* spp.; and marshes connected to the Aguapey river are interspersed with tall grasses of *Rhynchospora corymbosa* and Panicum spp. In addition, a small proportion of riparian natural forest patches remain along the Aguapey river [13]. Until the recent development of tree plantations, the Aguapey basin was mainly managed for extensive cattle ranching under natural pastures on large private properties (from 1000 to 20,000 hectares), while a minor proportion was utilized as croplands [10]. Thus, primary use of this landscape for extensive cattle grazing has facilitated the survival of one of the most extensive and biodiverse natural grassland areas in Argentina [3]. In this regard, this area was identified as an Important Bird Area where eight globally threatened (Endangered (EN), Vulnerable (VU)) and three 'Near Threatened' (NT) grassland bird species reside, according to the IUCN Red List Category (EN: *Xanthopsar flavus*, *Sporophila palustris*, *S. zelichi*; VU: *S. cinnamomea*, *Culicivora caudacuta*, *Alectrus risora*, *Xolmis diminicanus*, *Anthus nattereri*; NT: *Rhea americana*, *S. ruficolis*, *S. hypochroma*) [30]. It is also habitat of one of the last populations of the Pampas deer (*Ozotoceros bezoarticus*), a locally endangered and globally NT deer species dependent on natural grasslands [31]. Furthermore, the Aguapey basin also has a unique cultural value given that it still preserves traditional "gaucho" cattle ranching practices (which date from as far back as the 17th century), which is key to preserve the remaining areas of natural grasslands [3,32]. Despite the clearly important cultural and ecological value of this basin, it lacks formal protection [3].

The Aguapey VGA is distributed across two districts: Ituzaingó, towards the north of the study area, and Santo Tomé in the southern region (Figure 1c). Over the past decades, given the increased demand for forest-derived products such as cellulose, sawdust, and fiber boards, and the Government's support for the forest industry, both Ituzaingó and Santo Tomé districts have experienced a rapid and expansive development of tree plantations [33].

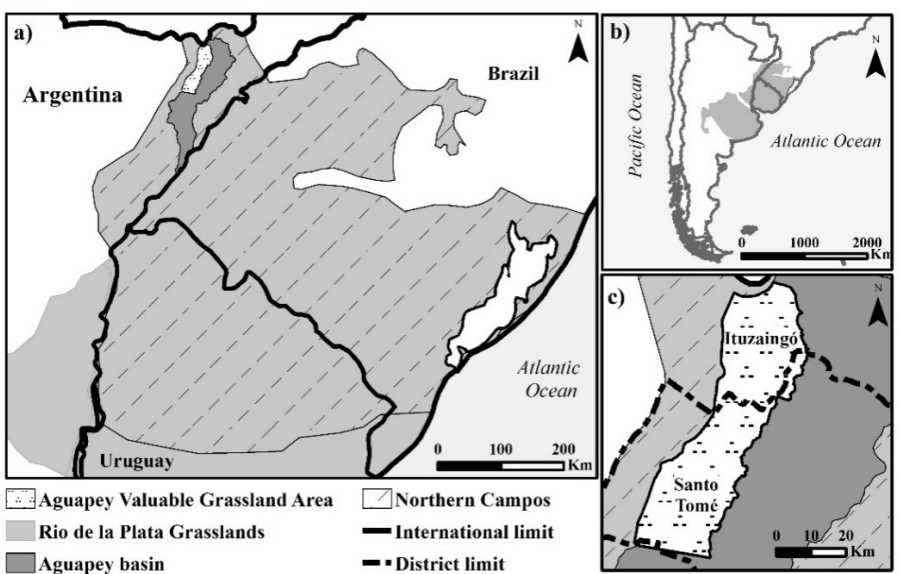

**Figure 1.** Study area: (**a**) Geographic location of the Aguapey VGA, within the Aguapey basin in the Northern Campos region of the Rio de la Plata Grasslands; (**b**) geographic location of the Rio de la Plata Grasslands; (**c**) study regions per district.

### 2.2. Satellite Data Collection and Pre-Processing

Remote sensing has been widely recognized as the most significant technology for effectively mapping the land cover units within a landscape. This is due to its numerous advantages compared to field-based assessments, such as study-area scale factors, cost-effectiveness, and the repeatability of observations [34,35]. Furthermore, remote sensing images offer an extraordinary capability to obtain past, present, and future land cover patterns to elucidate land cover change analysis [36]. Among many satellite-based Earth observation programs, the NASA Landsat program has been widely used for land cover change analyses due to it being the longest uninterrupted program since 1972, freely providing global coverage data at moderate spatial and temporal resolution (30 m, 16 days revisit time, respectively) [37]. Landsat Thematic Mapper (TM) and Landsat Operational Land Imager (OLI) Level 2 images (Path/Row 225/079), obtained from the Earth Explorer USGS public domain service (https://earthexplorer.usgs.gov; accessed on 17 March 2020), were therefore used to characterize the Aguapey VGA's land cover across four annual periods (TM: 1999–2000, 2006–2007; OLI: 2014–2015, and 2019–2020) following a phenological approach (see Section 2.3). In this regard, each period includes monthly Landsat images beginning in the southern hemisphere's winter (June/July) until the following autumn (April/May), encompassing a complete growing season of Aguapey VGA's natural grasslands [38]. The list of the satellite images used in this study, as well as the details of the geometric, radiometric, and atmospheric corrections are provided in Appendix A.

### 2.3. Land Cover Characterization

For each annual period, normalized difference vegetation index (NDVI) temporal series were created from monthly Landsat TM and Landsat OLI images in order to characterize the two main covers within the study area (grassland and tree plantation) following a phenological approach. The NDVI is calculated as

$$NDVI = (\rho NIR - \rho R)/(\rho NIR + \rho R), \tag{1}$$

where $\rho R$ is the reflectance in the red region of the electromagnetic spectrum (~0.68 μm, Band 3 and Band 4 of Landsat TM and Landsat OLI, respectively) and $\rho NIR$ is the reflectance in the near infrared range (~0.8 μm, Band 4 and Band 5 of Landsat TM and Landsat OLI, respectively). The NDVI is a linear estimator of the fraction of absorbed pho-

tosynthetically active radiation intercepted by vegetation (fAPAR) and is highly correlated with structural and functional attributes of the vegetation, such as the aboveground net primary productivity [39]. Land cover classifications based on multi-temporal NDVI data have the advantage of reducing the dimensions of the spectral data, which allows derivation of spectral signatures that are easy to interpret in biological terms [40]. In addition, among many other spectral vegetation indices, the NDVI has been proved to effectively detect seasonal and inter-annual changes in vegetation growth and activity, particularly at low or moderate vegetation amounts such as in grassland areas [41,42]; however, it tends to saturate under high biomass conditions and is very sensitive to canopy background variation [41]. In spite of its limitations, previous studies across the RPG region have shown that monthly NDVI time series data effectively discriminate temperate grasslands from other cover types (e.g., croplands, forested areas) based on their unique phenological characteristics [16,24,40,43,44]. It was therefore preferred in this scenario to other available spectral indices.

For each period, the NDVI datasets were stacked and resampled to 30 m pixel size using the nearest neighbor method in order to preserve the original image radiometric information. Given that ground-truth information was not available for annual periods prior to 2014, unsupervised classifications were performed to each annual NDVI dataset using the ISODATA algorithm (following a similar approach that was previously performed for the RPG region; see [16,43]. The ISODATA algorithm was set to generate a maximum of 20 classes using 100 iterations, a tolerance threshold of 5%, and maximum standard deviation of 1. Phenological signatures were built for each output class and period: (i) the classes that registered NDVI values that ranged between 0.4 and 0.7 and followed a unimodal response, with the lowest NDVI values in winter (July–August–September), a steady increase until the summer months (December–March) and afterwards a slow NDVI decrease, were identified as 'temperate grasslands' [38], (ii) the classes that registered NDVI values above 0.7 throughout the year were identified as 'tree plantation' [15], and (iii) the classes that did not show a clear phenological pattern to be included into any of the previous categories were identified as 'other' covers (see Appendix B for further details of the land covers' phenological signatures).

Two different procedures were followed to assess classification accuracy based on the availability of ground truth data. For the period 2014–2015, ground truth information was available from the 2014 Aguapey VGA's land cover thematic map [12]; therefore, a 1000 pixel-based contingency matrix was generated [45]. The remaining periods (1999–2000, 2006–2007, and 2019–2020) were not supported with available ground truth data. Therefore, adopting a similar approach to that implemented previously by [44,46], 200 pixels were randomly distributed across the study area for each remaining period; the pixels' ground truth land cover were visually interpreted using both high resolution Google Earth imagery and spectral signatures. A 200 pixel-based contingency matrix was used to assess the unsupervised classifications' accuracies for each remaining period. Finally, a post-classification majority filter (7 × 7 pixels) was applied in order to smooth the 'salt and pepper' appearance of the resulting classifications and improve their quality through the elimination of spuriously classified pixels [40]. For image processing the software ENVI, Version 5.5 was used.

### 2.4. Landscape Structure and Dynamics

A set of six traditional landscape metrics were estimated for each annual period to quantify fundamental aspects of Aguapey's VGA land cover composition and grassland spatial configuration (Table 1). The spatial configuration metrics chosen have been shown to be useful descriptors of landscape fragmentation [24,47,48], and it is proposed that these metrics can capture the complexity of the spatial arrangement of the patches without providing redundant information [49]. In this regard, the effective mesh size (EMS) has been proposed as one of the most relevant measures of the degree of landscape fragmentation, given that it simultaneously considers the patch size and the level of dissection [22]. It

has also been frequently used in a wide range of ecosystems and regions [24,48]. The remaining spatial configuration metrics (PD, PS, SHAPE, ENN, ECON) were used as complementary fragmentation measures to describe more specific attributes of grassland configuration [16,47,48,50]. The landscape metrics were assessed using FRAGSTATS 4.2 [51]; the patch level metrics were averaged for each annual period to statistically assess the difference between the means using Kruskal Wallis analyses, given the nonparametric nature of the variables (see Appendix C).

In order to gain a more detailed and local-scale understanding of the grassland afforestation process which occurred between 1999 and 2020 in the Aguapey VGA, a further, supplementary analysis of grassland loss was undertaken in the two districts within the study area: Ituzaingó and Santo Tomé. This was undertaken to ensure that more local differences were also considered and understood within the analysis and to uncover potential differences at the local policy level.

**Table 1.** Landscape metrics selected to better reflect the Aguapey VGA's composition and the grassland fragmentation.

| Metric | Units | Level of Analysis | Explanation |
|---|---|---|---|
| **Landscape Composition** | | | |
| Percentage of Landscape (PLAND) | % | class | Considered as the most important and useful information to describe a landscape [52]. |
| **Grassland Spatial Configuration** | | | |
| Effective Mesh Size (EMS) | $km^2$ | class | Quantifies the probability that two randomly chosen points in a study area are connected [53]. It is not sensitive to the omission or inclusion of small patches and has a monotonous response through to different fragmentation stages. The greater the effective mesh size, the lower the fragmentation level [22,53]. |
| Patch Density (PD) | n° per 100 hectares | class | It is a simple measure of the degree of subdivision of a cover type; however, it presents a unimodal relationship with the amount of disturbance leading to possible misinterpretations [24,48,52]. |
| Patch Size (PS) | $km^2$ | patch | Progressive reduction in the patch area is a key component of ecosystem fragmentation [52]. |
| Shape Index (SHAPE) | unitless | patch | Measure of the overall patch shape complexity. SHAPE would be 1 when the patch is square (simple geometry) and would tend to increase with increasing shape complexity. Higher index values indicate higher fragmentation due to disturbances on the edges of an ecosystem [52]. |
| Euclidean Nearest Neighbor Distance (ENN) | m | patch | Measure of patch isolation estimated as the shortest straight-line distance between the focal patch and its nearest neighbor increases in ENN could be indicative of higher ecosystem fragmentation [47]. |
| Edge Contrast Index (ECON) | % | patch | Measure of the relative contrast along the patch's perimeter with its surroundings. ECON would be zero when the patch perimeter has no contrast with its surroundings, and it would tend to increase with increasing contrast between cover types [52]; higher index values suggest higher levels of landscape fragmentation [50]. Contrast levels were based on the structural differences between grassland and the remaining land covers, and therefore set to 1 for tree plantation and to 0.25 for other covers. |

### 2.5. Land Covers' Transition Probability Analysis

In order to gain further understanding regarding the change dynamics between land covers along the four annual periods, relative transition probabilities were derived for each land cover from the thematic maps [16,54]. Transition probabilities account for the proportion of one cover making a transition to another cover between any two points in time; therefore, they provide an indication of how much change happens to a particular land cover over a time period rather than an overall measurement of areal land cover. To compute the transition probabilities, the land cover thematic maps were vectorized and intersection maps were created from consecutive periods (1999–2000 and 2006–2007, 2006–2007 and 2014–2015, 2014–2015 and 2019–2020). Transition probabilities from land cover $i$ in time $n$ to land cover $j$ in time $n + 1$, $p_{ij}(n, n + 1)$, were estimated as

$$p_{ij}(n, n + 1) = A_{ij}(n + 1) / A_i(n), \tag{2}$$

where $A_i(n)$ is the area occupied by the landcover class $i$ in the first period selected and $A_{ij}(n + 1)$ is the area that changed from landcover class $i$ to $j$ in a later period.

## 3. Results

### 3.1. Land Cover Characterisation

Unsupervised classification of the Aguapey VGA study region reached overall accuracy values exceeding 85% for all periods studied. Both grasslands and tree plantations were effectively discriminated from each other as well as from other cover classes, showing producer and user accuracy values above 80% (Table 2, Appendix D). On the other hand, other covers' classification accuracy varied between periods. In 1999–2000 and 2006–2007, other covers showed moderate to high producer and user accuracy values, whereas from 2014 onwards, these values were below 50%. This indicates a high level of misclassification, in particular, with grassland cover (Table 2, Appendix D). Overall, the high accuracy values achieved, mainly for grasslands and tree plantations, provided enough confidence regarding their classification across the study area and therefore for the following analyses. The results associated to other covers were carefully interpreted given their high level of misclassification found in the last two annual periods (2014–2015 and 2019–2020).

**Table 2.** Producer and user accuracy values for the land cover classes and overall accuracy values for each annual period under study. See Appendix C for a detailed analysis of the contingency matrices.

| | Grassland | | Tree Plantation | | Other Covers | | |
|---|---|---|---|---|---|---|---|
| **Annual Period** | **Producer** | **User** | **Producer** | **User** | **Producer** | **User** | **Overall** |
| 1999–2000 | 97.51 | 92.9 | 80.0 | 100 | 64.71 | 81.48 | 91.5 |
| 2006–2007 | 95.56 | 92.81 | 83.33 | 100 | 81.13 | 84.13 | 91.0 |
| 2014–2015 | 90.85 | 91.24 | 98.67 | 82.59 | 15.09 | 22.86 | 88.6 |
| 2019–2020 | 96.22 | 87.5 | 98.43 | 98.08 | 10.53 | 50.0 | 89.5 |

### 3.2. Landscape Structure and Dynamics

Considerable changes in land cover occurred on the Aguapey's VGA over the last 20 years (Table 3; Figure 2). Between 1999 and 2020, the total cover of grasslands decreased almost 22% (from 1434.81 km$^2$ to 1083.83 km$^2$, a relative decrease of nearly 25% of its original cover), while the total coverage of tree plantations increased nearly 26% (from 59.35 km$^2$ to 476.05 km$^2$, a total relative increase of 702.11%) (Table 3; Figure 2).

Most of the grassland loss occurred between 1999–2000 and 2006–2007 when the total grassland coverage reduced by ~20% (from 89.78% to 70.04%). However, between these first two periods, tree plantation total coverage increased only ~3% (from 3.71% to 6.72%) (Table 3; Figure 2a,b); other covers, on the contrary, registered a rise of almost 17% (from 6.5% to 23.24%) occupying formerly grassland area (Table 3; Figure 2a,b). From 2006–2007 to 2014–2015, grassland cover remained relatively unchanged while tree plantation in-

creased by ~21% (from 6.72% to 27.52%, its largest change between periods), almost the same proportion was recorded for other covers' reduction (from 23.24% to 2.2%) (Table 3; Figure 2b,c). From 2014–2015 to 2019–2020, all land covers registered subtle changes (Table 3; Figure 2c,d).

**Table 3.** Area of each land cover class for the Aguapey VGA in four annual periods based on the unsupervised classification thematic maps.

| Annual Period | Grassland %(km$^2$) | Tree Plantation %(km$^2$) | Other Covers %(km$^2$) |
|---|---|---|---|
| 1999–2000 | 89.78 (1434.81) | 3.71 (59.35) | 6.5 (103.93) |
| 2006–2007 | 70.04 (1119.34) | 6.72 (107.33) | 23.24 (371.44) |
| 2014–2015 | 70.28 (1123.19) | 27.52 (439.72) | 2.2 (35.2) |
| 2019–2020 | 67.82 (1088.83) | 29.79 (476.06) | 2.39 (38.28) |
| Relative change in specific land cover (%) [1] | −24.46 | +702.11 | |
| Total change within the Aguapey VGA (%) [2] | −21.96 | +26.08 | |

[1] Relative change in specific land cover's area between 1999–2000 and 2019–2020 was calculated as $\Delta$ *relative* = 100 × ($Y_{final} - Y_{initial,}$)/$Y_{initial,}$, where $Y$ is the area of the land cover type. [2] Total change in area between 1999–2000 and 201–2020 within the Aguapey VGA was calculated as $\Delta$ total = 100 × ($Y_{final} - Y_{initial}$)/Aguapey VGA *area*.

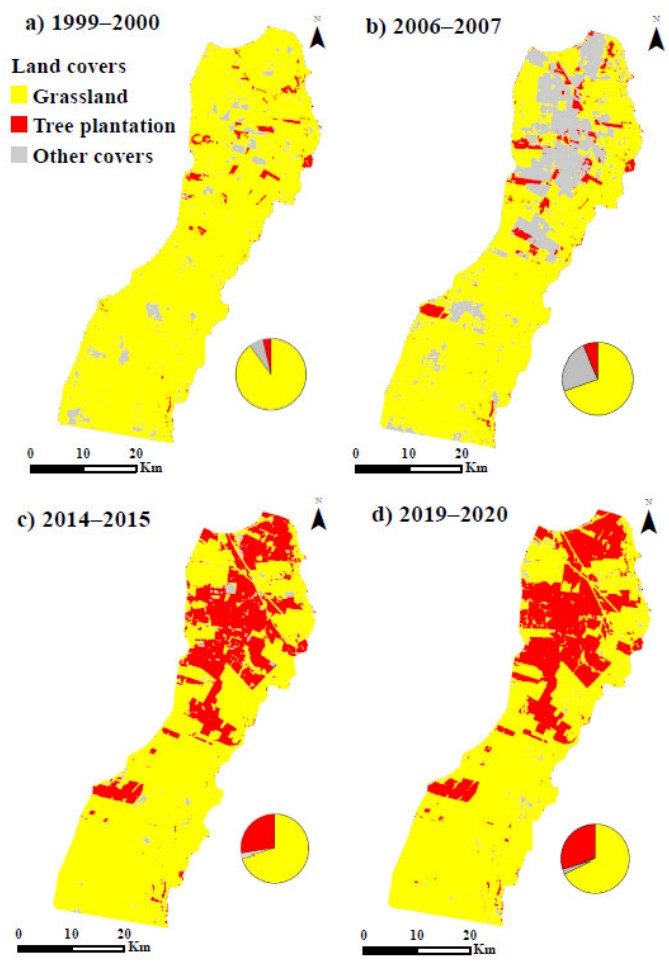

**Figure 2.** Land cover thematic maps for (**a**) 1999–2000, (**b**) 2006–2007, (**c**) 2014–2015, and (**d**) 2019–2020, based on unsupervised classifications of NDVI time-series datasets derived from Landsat TM (**a**,**b**) and Landsat OLI images (**c**,**d**). Pie charts indicate the percentage of each land cover type for each period.

The transition probability analysis provided further evidence regarding the land cover change dynamics along the 20 years-period studied (Figure 3). The grassland's

largest transition to a different land cover took place between 1999–2000 and 2006–2007, when 22% (315.49 km²) of the original grassland cover changed into 'other covers' and 5% (71.7 km²) changed into tree plantation (73% of grassland, 1046.86 km², remained as grassland) (Figure 3a). During the following periods, increasing proportions of grassland areas were preserved as grasslands; while 86% was preserved between 2006–2007 and 2014–2015, which represented a grassland area of 62.63 km², 92% (1033.33 km²) was preserved from 2014–2015 to 2019–2020 (Figure 3b,c). Furthermore, and contrary to what was observed during the first period, in those that followed, the proportion of grassland that changed into tree plantation surpassed the proportion that changed into other covers: from 2006–2007 to 2014–2015, 12% (134.32 km²) changed into tree plantation and 2% (22.39 km²) changed into other covers; from 2014–2015 to 2019–2020, 5% (56.16 km²) changed into tree plantation and 3% (33.7 km²) changed into other covers (Figure 3b,c). Similarly to grasslands, tree plantation recorded the largest proportion of transition during the first period (23% of tree plantation's original cover, which represented 13.47 km², changed into grassland and 11%, 6.44 km², changed into other covers) and, from 2007 onwards, increasingly larger proportions of tree plantation remained as tree plantation (84% and 92%, representing 90.16 km² and 404.54 km², respectively) (Figure 3a–c). Finally, other covers registered the largest proportions of transition into both tree plantation and grassland along the 20 years-period, in comparison with the remaining land covers (Figure 3). Between 1999–2000 and 2006–2007, almost 60% (58.8 km²) of the initial other covers' area changed into grassland, while 40% ((41.57 km²) remained as other covers and only 3% (3.09 km²) changed into tree plantation (Figure 3a). However, from 2006 to 2007 onwards, almost all of the remaining other covers' area was transformed mainly into tree plantation (58% between 2006–2007 and 2014–2015, representing an area of 215.44 km²; 50%, 17.6 km², was transformed between 2014–2015 and 2019–2020), and to a lesser extent, into grassland (38%, equivalent to 141.15 km² was transformed between 2006–2007 and 2014–2015, 44%, 15.5 km², between 2014–2015 and 2019–2020) (Figure 3b,c).

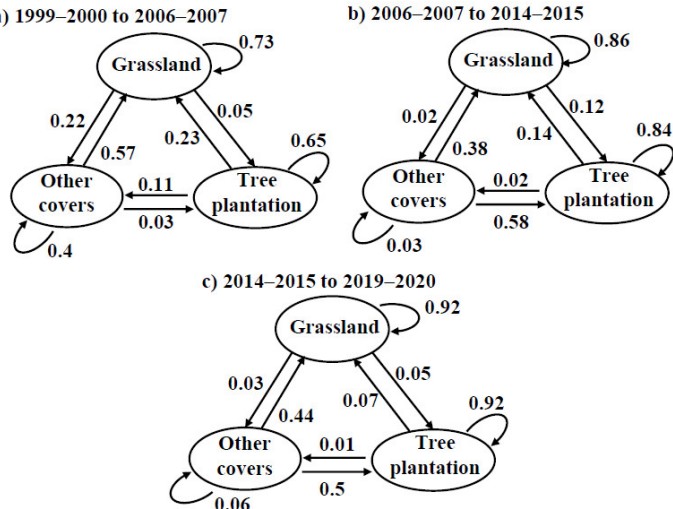

**Figure 3.** Land covers' transition probability analysis within the Aguapey VGA between (**a**) 1999–2000 and 2006–2007, (**b**) 2006–2007 and 2014–2015, (**c**) 2014–2015 and 2019–2020.

The transition probability analyses suggest that the conversion of grassland into tree plantations was a progressive process of change indicated by an initial main conversion of grasslands into 'other covers' (from 1999–2000 to 2006–2007) (Figures 2a,b and 3b), which in turn were converted largely into tree plantations (from 2007 onwards) Figures 2b–d and 3b,c). Figure 4 provides an example of this grassland–other covers–tree plantations' transition along the 20-year period studied and reveals the changes exhibited in the phenological cycle through the years. In this regard, it is possible to see a decrease in the NDVI values from 1999–2000 to 2006–2007 given the change from grassland to other covers and, from

2006–2007 onwards, an NDVI increase due to the transition from other covers to established tree plantations (see also Appendix B). Based on these results and considering that the growth period of *Eucalyptus* spp. and *Pinus* spp. range between 4 to 10 years [32], other covers were in the majority considered as the initial stages of tree plantations, comprising bare soil with young sapling tree species and grasses. This would explain the low NDVI values registered for other covers, particularly between 2006 and 2007, and define the different phenological behavior exhibited between periods given the possible changes regarding amounts of bare soil, grass, or young tree stands, which would also explain the high transition values recorded from other covers to tree plantation and grassland. Furthermore, the high increase of tree plantation cover recorded mainly in 2014–2015 (Table 3) could be explained by the progressive maturation of the other covers into tree plantations from 2007 onwards (Figure 3b,c).

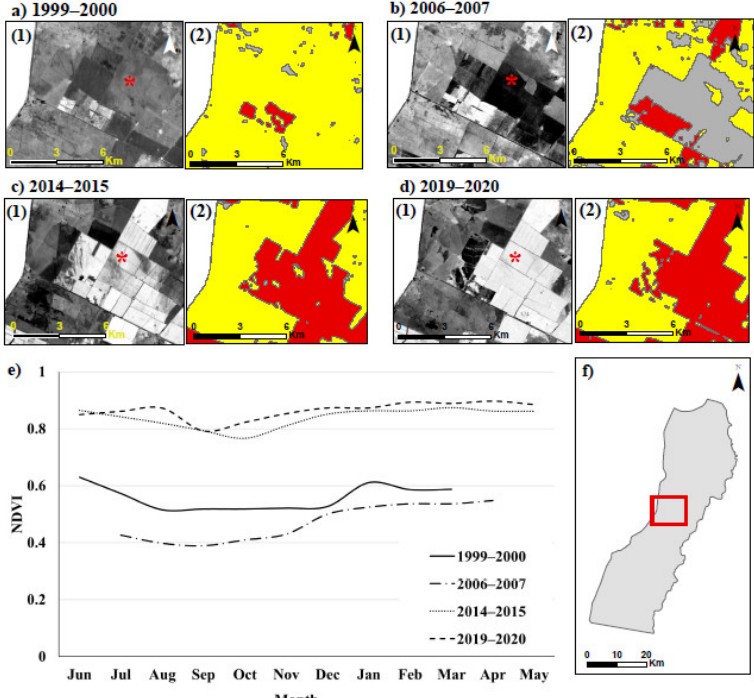

**Figure 4.** Example of transition between grassland–other covers–tree plantation from 1999 to 2020 in an area located at the central region of the Aguapey VGA (**a–d**), using as base maps (1) NDVI images from November (the only month included in all annual periods) and (2) unsupervised classification images with yellow showing grassland, grey other covers, and red tree plantation; (**e**) phenological signatures of a particular region of interest (*) for each annual period studied; missing values within the phenological cycle were estimated as the average between the predecessor and the following values; (**f**) Aguapey VGA study area.

### 3.3. Grasslands Spatial Configuration

Over the 20 year-period studied, concordantly with the decrease in Aguapey VGA's grassland cover, grasslands also displayed an increased level of fragmentation (Table 4, Figure 5). In 1999–2000, grasslands registered the lowest level of fragmentation (EMS 1281.38 km$^2$, Table 4); during this period, grasslands were characterized with a low density of patches (PD 0.05, Table 4) that were on average significantly larger (PS 17.07 km$^2$, Figure 5a) and significantly less irregular (SHAPE 1.31, Figure 5b) than the patches from the other studied periods. From 1999–2000 to 2006–2007 there was an increased level of grassland fragmentation (EMS 608.27 km$^2$, Table 4), accompanied by an increased density of grassland patches of significantly lower average size and higher complexity (PD 0.26, Table 4; PS 2.68 km$^2$, Figure 5a; SHAPE 1.39, Figure 5b; respectively). No significant

differences were observed in the measure of distance between patches nor contrast between patches and their surroundings during the first two periods (ENN ranged between 147.95 m in 1999–2000 and 161.68 m in 2006–2007, Figure 5c; ECON ranged between 22.31% in 1999–2000 and 27.03% in 2006–2007, Figure 5d). In 2014–2015, the Aguapey VGA's grasslands registered the highest level of fragmentation (EMS 510.35 km$^2$, Table 4), which was mainly explained by a decreased density of grassland patches (PD 0.14, Table 4) and a significant increase in both the distance between patches (ENN 235.28 m, Figure 5c) and the contrast between grassland patches and their surroundings (ECON 73.14%, Figure 5d). No differences were observed regarding the size and shape of grassland patches with respect to the former periods (Figure 5a,b, respectively). Finally, from 2014–2015 until 2019–2020, grassland cover remained relatively unchanged: a slight recovery in the level of fragmentation was exhibited (EMS 540.95 km$^2$, Table 4), accompanied by a decrease in density of patches (PD 0.12, Table 4). No differences were observed in the average size, shape, distance, and edge contrast between grassland patches from the two later periods (Figure 5a–d, respectively).

**Table 4.** Grassland landscape metrics at class level (PD, EMS).

| | Grasslands | |
|---|---|---|
| **Annual Period** | **PD** | **EMS (km$^2$)** |
| 1999–2000 | 0.05 | 1281.38 |
| 2006–2007 | 0.26 | 608.27 |
| 2014–2015 | 0.14 | 510.35 |
| 2019–2020 | 0.12 | 540.95 |

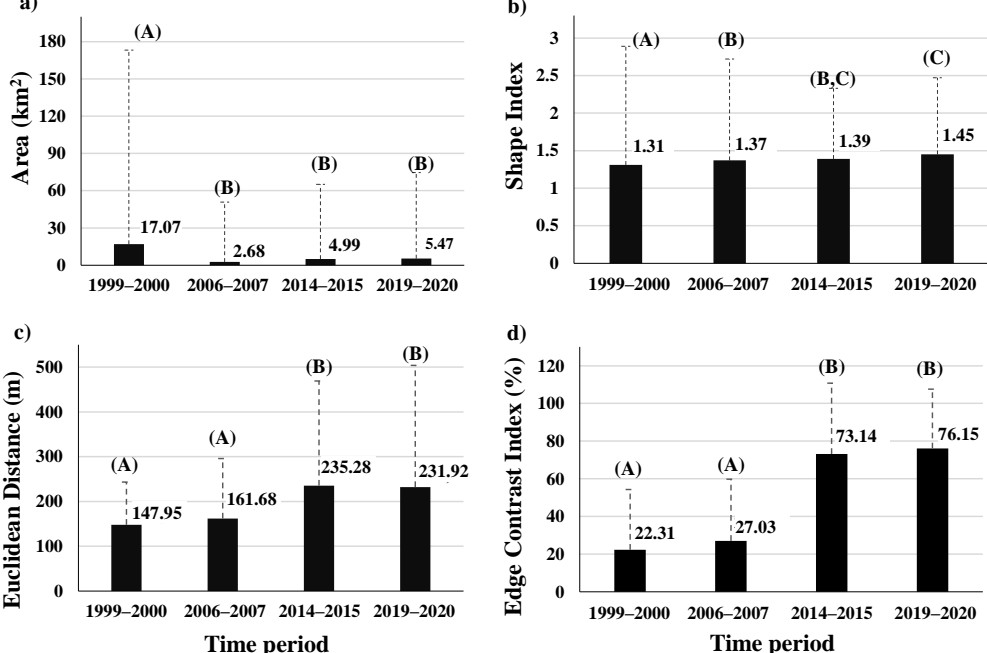

**Figure 5.** Mean values for each grassland landscape metric at patch level, (**a**) PS, (**b**) SHAPE, (**c**) ENN, (**d**) ECON; vertical bars represent the standard deviation. Significant differences between time periods based on Kruskal Wallis test is represented by different letters ((A), (B), (C)) (see Appendix C for the Kruskal Wallis tests results).

*3.4. District Scale Analysis*

In order to gain a deeper understanding of the afforestation process within the study area, spatio-temporal changes in land cover composition were also analyzed at district scale (Table 5). This analysis showed that during the first annual period (1999–2000) natural

grasslands were the main land cover within both Ituzaingó and Santo Tomé, occupying 85.1% (549.95 km$^2$) and 93.15% (878.14 km$^2$) of the district area, respectively. However, between 1999 and 2020, Ituzaingó's grasslands suffered a relative decrease of slightly more than 50%, whereas Santo Tomé's grasslands remained relatively unchanged (relative decrease of 6.47%) (Table 5). A detailed analysis per period showed that the major loss of total grassland area in Ituzaingó occurred between 1999–2000 and 2006–2007 (from an initial area of 549.95 km$^2$ to an area of 323.01%), resembling the decreasing trend observed for the entire Aguapey VGA landscape. Furthermore, over the 20-year study period, Ituzaingó's grassland cover decreased nearly 290 km$^2$, representing almost 83% of the total loss of grassland cover within the Aguapey VGA (see Table 3). In 2019–2020, 75.74% of the remaining Aguapey VGA grassland was distributed in the Santo Tomé district (820.13 km$^2$, Tables 3 and 5).

**Table 5.** Area of each land cover class for the Aguapey VGA's districts in four annual periods based on the unsupervised classification thematic maps.

| | Ituzaingó | | Santo Tomé | |
|---|---|---|---|---|
| **Annual Period** | **Grassland %(km$^2$)** | **Tree Plantation %(km$^2$)** | **Grassland %(km$^2$)** | **Tree Plantation %(km$^2$)** |
| 1999–2000 | 85.1 (552.92) | 6.91 (44.67) | 93.15 (881.11) | 1.48 (13.91) |
| 2006–2007 | 49.98 (325.03) | 10.74 (69.46) | 83.92 (793.21) | 3.9 (36.76) |
| 2014–2015 | 44.68 (292.75) | 51.46 (332.54) | 87.55 (829.35) | 11.27 (106.2) |
| 2019–2020 | 39.95 (262.75) | 56.92 (367.87) | 86.51 (820.13) | 11.37 (107.23) |
| Relative change in specific land cover (%) | −52.48 | +723.52 | −6.92 | +670.88 |
| Total change within district (%) | −44.9 | +50.0 | −6.47 | +9.9 |

Conversely to the loss of grassland, tree plantation cover increased at both Ituzaingó and Santo Tomé between 1999 and 2020; however, Ituzaingó registered the largest total coverage increase within the district (50% Ituzaingó, 9.9% Santo Tomé) (Table 5). A detailed analysis per period exhibited that Ituzaingó's tree plantations increased markedly before 2007, becoming the dominant land cover at the end of the 20-year study period (total coverage of 56.92%, Table 5). In addition, from 1999 to 2020, the original cover of tree plantations in Ituzaingó increased 323.2 km$^2$, an increment that represented almost 70% of the tree plantation expansion within the Aguapey VGA landscape (see Table 3).

In addition, the landscape metric analysis indicated that the level of Ituzaingo's grassland fragmentation steadily increased over the 20 years period (the EMS declined from 466.22 km$^2$ in 1999–2000 to 34.71 km$^2$ in 2019–2020) (Table 6). From 1999–2000 until 2014–2015, the increasing grassland fragmentation within Ituzaingó could be explained by a rise in the number of grassland patches (PD increased from 0.06 to 0.53 between the first two periods and then decreased to 0.24) (Table 6) which were on average significantly smaller than the grassland patches registered in 1999–2000 (Figure 6a) and more irregular (SHAPE significantly increased from 1.35 to 1.47) (Figure 6b). Furthermore, from 2015 onwards, there was a significant increase not only in the distance between Ituzaingó's grassland patches (Figure 6c) but also in their contrast with the surroundings (Figure 6d).

**Table 6.** Grassland landscape metrics at class level (PD, EMS) for the Aguapey VGA districts.

| | Grasslands | | | |
|---|---|---|---|---|
| | Ituzaingó | | Santo Tomé | |
| **Annual Period** | **PD** | **EMS (km$^2$)** | **PD** | **EMS (km$^2$)** |
| 1999–2000 | 0.06 | 466.22 | 0.05 | 814.24 |
| 2006–2007 | 0.53 | 53.97 | 0.08 | 655.97 |
| 2014–2015 | 0.29 | 39.94 | 0.04 | 691.26 |
| 2019–2020 | 0.24 | 34.71 | 0.05 | 675.72 |

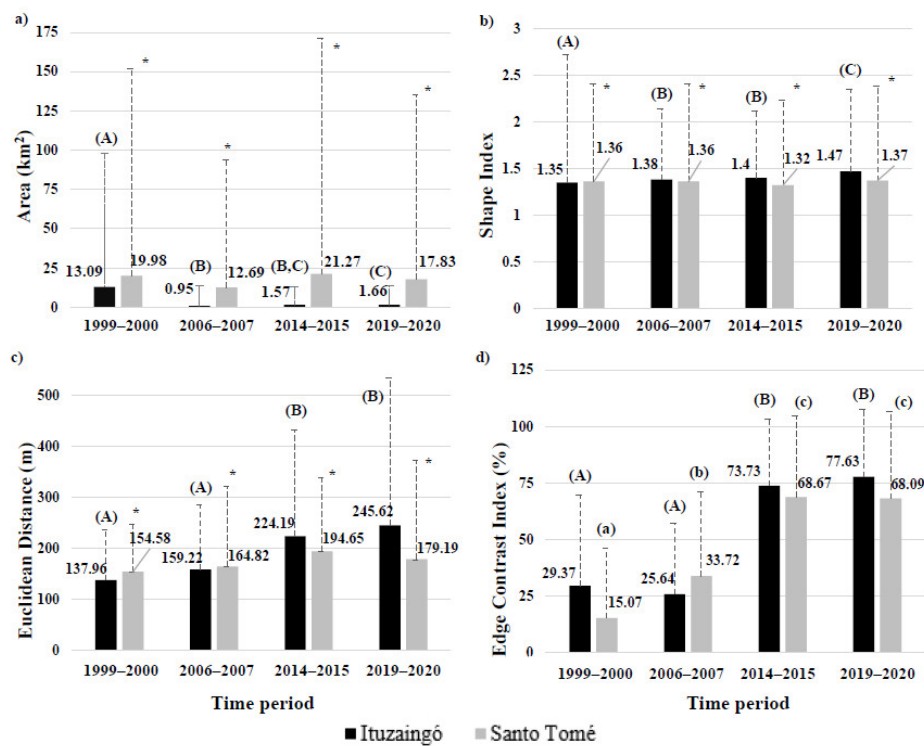

**Figure 6.** Mean values for grassland landscape metric at patch level, (**a**) PS, (**b**) SHAPE, (**c**) ENN, (**d**) ECON, per district (Ituzaingó, black bars; Santo Tomé, gray bars); vertical bars represent the standard deviation. Significant differences between time periods based on Kruskal Wallis test is represented by different letters ((A), (B), (C), Ituzaingó district; (a), (b), (c), Santo Tomé district); no significant differences between time periods based on Kruskal Wallis test is represented by * (see Appendix C for the Kruskal Wallis tests results).

Conversely, although the Santo Tomé district also showed increasing levels of fragmentation over the 20 years period studied, the maximum fragmentation was registered in 2006–2007 (EMS 655.97 km$^2$) (Table 6). The increasing fragmentation within the district could be explained by an increment in patch density between 1999–2000 and 2006–2007 (PD changed from 0.05 to 0.08, respectively) (Table 6), as well as by the higher contrast between Santo Tomé's grassland patches and their surroundings registered from 1999–2000 until 2014–2015 (Figure 6d). Contrary to what was observed in Ituzaingó, no significant differences were observed in the PS, SHAPE nor ENN values of Santo Tomé's grasslands between periods (Figure 6a–c).

## 4. Discussion

The research presented in this study addresses the analysis of the spatio-temporal changes manifesting in the landscape structure of one of the globally most well-preserved temperate grassland areas, the Aguapey VGA, Argentina. The study is conducted between 1999 and 2020 in a location considered an internationally important temperate grassland area, and a refuge for more than 10 globally endangered grassland species. The study findings assert that, during this study period, structural changes within the Aguapey VGA were principally driven by the expansion of exotic tree plantations (*Eucaliptus* spp. and *Pinus* spp.). Overall, the analyses indicate that a vast expansion of the forestry activity occurred, and that this expansion was at the expense of temperate grassland, which suffered area loss and fragmentation. This loss was temporally consistent with the establishment of the national afforestation policy enacted in the mid-1990s and the Aguapey region's suitability for the development of this forestry activity [11,12]. In this regard, between 1999 and 2020, a total of 350.98 km$^2$ (≈22%) of the Aguapey VGA's grassland were lost while tree

plantation cover increased from 59.35 km$^2$ to 476.05 km$^2$ (almost 26% of the Aguapey VGA). Despite this significant land cover change, in 2019–2020 temperate grasslands remained the majority land cover of the Aguapey VGA, occupying almost 68% of the area. Its continuing dominance highlights its ongoing significance as a refuge for this internationally important grassland system that has characterized the RPG region for millennia prior to European colonization [55].

A detailed analysis of different landscape metrics, together with the assessment of the transition proportion between land covers, revealed a complex transition dynamic between grassland and tree plantations that led to different fragmentation stages of temperate grasslands over the 20-year study period. Between 1999 and 2007, the loss of grassland cover, together with an increased density of patches of lower size and higher shape complexity, indicated the perforation process of the grassland [22]. This was driven mainly by the conversion of grasslands into areas of bare soil and occupied by young tree stands that are spectrally characterized very differently to established forest vegetation. From 2006 to 2015, the decreased grassland patches, which were located at a higher distance from each other, and which showed higher structural contrast with the land cover that surrounded them, revealed that the grassland cover was going through an attrition process [22,24,56], evidencing that tree plantations were establishing themselves during this period, and highlighting an increase in the intensity of the fragmentation process that had started in 1999. Contrary to the previous periods, from 2015 to 2020 a slight recovery noted by the reduced level of grassland fragmentation was recorded and supported by the decrease in patch density.

The previous analysis identified that the largest afforestation impacts in the region occurred between 1999 and 2006; and more specifically comprised of the perforation of the original grassland cover by emerging tree plantation stands (typically in their initial growth phases with young tree species and grasses coexisting). The identification of the early development of tree plantations between 1999 and 2007 places a spotlight on the contemporary move to forest activity; and provides an evidence base to suggest that the grassland loss and fragmentation was motivated by the national economic incentives for the development of forestry activity from the late-1990s onwards. However, the new analysis undertaken in this paper highlights that between 2007 and 2020 grassland cover remained relatively unchanged. This is potentially noteworthy as even during this later period the national afforestation policy remained in place; yet the apparent uptake by landowners was reduced.

The differences in the afforestation dynamics in the temporal period (1999–2020), highlighted in the current study, could indicate that other factors, in addition to the national policy, were influencing the structural changes across the Aguapey VGA. In this regard, it is pertinent to consider that the cattle ranching practice in the region is a tradition deeply rooted in the local culture [32]. In 2006, local producers within the Campos region, together with BirdLife International and various national NGO's, started working on a regional initiative, the "Southern Cone Grasslands Alliance" (www.birdlife.org/americas/programmes; accessed on 8 August 2020). This initiative was created to enhance traditional cattle ranching practices and preserve the temperate grassland whilst also sustaining this environmentally sound economic activity [57]. In addition, in 2012, the Alliance launched the project 'Incentives for conservation of natural grasslands in the Southern Cone' which provided financial support for local producers; and most significantly the Aguapey VGA was selected as one of the pilot sites for this project initiative. Therefore, the increasing interaction between local NGO's and producers (with a common interest in continuing traditional rearing cattle practices), combined with the financial support for local farmers, may have interrupted the trend of exotic tree plantation expansion within the Aguapey VGA from 2007 onwards.

The analysis performed at district scale has revealed that the afforestation process within the Aguapey VGA took place mainly in the Ituzaingó district, where the loss of grassland amounted to more than 50% of the original cover; the evidence base shows that

this loss in grassland was directly relative to an increase of approximately 700% of tree plantations in 20 years. Conversely, the Santo Tomé grassland cover remained relatively unchanged between the periods studied, and by 2019–2020 grassland was still the main land cover occupying ~87% of the district area. Considering that both Ituzaingó and Santo Tomé share the same environmental and cultural characteristics within the Aguapey VGA, it is highly probable that the observed disparity between districts is influenced by different factors at a local scale. For example, 65% of the Ituzaingó district is covered by the Iberá Natural Reserve (next to the west of the Aguapey VGA), where the development of the forestry industry is highly restricted by law (Argentina's National Law 27.481). Therefore, a high concentration of tree plantations is expected in the remaining 35% of the district. Additionally, according to the First Forest Inventory of the Corrientes province [33], most tree plantations within Santo Tomé were distributed outside the Aguapey VGA towards the eastern region of the district with better edaphic conditions for the development of forestry activity. These findings, together with the different Aguapey VGA afforestation dynamics observed between periods, highlight that the afforestation process across this region was determined by a complex interplay of environmental, social, political, and economic factors, providing further evidence to support previous studies which suggest that landscape change is not a random process; rather, disproportionate changes occur in certain areas or periods given the influence of a wide array of factors [16,23,24,47].

The analysis presented in this study has clearly demonstrated that the afforestation process which occurred in the Aguapey VGA between 1999 and 2020 induced the significant loss of an important temperate grassland area (mainly within the northern region). Previous studies in the Campos region reported major costs to species diversity and ecosystem services due to the replacement of grassland ecosystems by tree plantations. For example, Phifer et al., (2016) [55] reported that Eucalyptus plantations reduced the richness and abundance of grassland-dependent bird species (such as *S. ruficollis* and *R. americana*) and their associated ecosystem services such as pest control, seed dispersal, and pollination. In addition, [29,58] registered that afforested temperate grasslands within the RPG led to localized water balance shifts which, in turn, triggered intense water and soil salinization processes, decreasing water and soil quality.

Furthermore, the substantial loss of temperate grassland systems has led to the isolation and segregation of many of the remnant grassland patches. The effects of habitat isolation on species' population viability have been extensively studied (see [19]); however, it has also been recognized that the fragmentation impacts are species-specific [59,60]. In this regard, [13] stated that six globally threatened bird species, distributed within the Aguapey basin, were highly impacted by the presence of tree plantations, which not only affected their breeding habitats but also impeded their ability to disperse between habitat patches. The impacts of temperate grassland fragmentation have also been reported in regard to the endangered Pampas deer [31]; however, a recent study suggested that Pampas deer could be positively selecting grassland patches within young tree plantations as refuge against predators [61]. The ecological effects are therefore complex and warrant further study. On this basis we propose that further species-specific analysis should be developed in this region to better understand the species-specific effects of afforestation practices.

In recent years, several international initiatives, such as the Bonn Challenge and the New York Declaration on Forests, have established ambitious targets for forest cover in-creases, which are seen as necessary to limit global warming by 2050 [62]. In order to achieve this ambitious goal the United Nations Framework Convention on Climate Change (UNFCCC) Clean Development Mechanism (CDM) promotes grassland ecosystems' afforestation as one of many global mitigation strategies [63], given the potential of forest industries to operate as a net sink for carbon [64,65]; furthermore, it was stated that afforestation projects have the most potential in developing countries due to the higher growth rates of forest and the land availability [66]. In this regard, Argentina has launched a new initiative to increase by 50% the tree plantation cover in the next nine years [27], overlooking the possible negative impacts that that the replacement of historic natural

temperate grasslands with exotic tree monocultures induces on species diversity and the ecosystem services previously mentioned.

Although grassland afforestation can increase carbon uptake by considerably increasing the aboveground biomass accumulation [15,67,68], this increment does not necessarily imply net long-term carbon sequestration [69]. For example, it has been observed that afforestation can produce a net loss of soil organic carbon as a consequence of different C allocation patterns between grasses and trees [70–72]; while other studies registered that a higher proportion of tree plantations' NPP can be lost by fire or appropriated through harvesting [73,74].

Natural grasslands soils represent very large carbon sinks at global scale [1,2]; therefore, where grassland ecosystems are preserved and sustainably managed, their largely belowground soil organic carbon stocks are secure from disturbances such as fire, deforestation, and disease [1,74,75]. Furthermore, a recent study reported that restoration of degraded grassland areas with high diversity of dominant grass species promoted high rates of soil carbon accumulation, increasing the ability of these areas to contribute to C sequestration [76]. Given that soils' natural grassland ecosystems could also play a key role in the carbon cycle, future studies should also be conducted which seek to assess the impact of temperate grassland loss and fragmentation on its role in carbon sequestration and to more critically examine the benefits of mixed land use to mitigate carbon emissions.

## 5. Conclusions

This study provides an insight into the impacts of the afforestation process between 1999 and 2020 in the Aguapey VGA. During the first 15 years, structural changes took place mainly within the northern region of the Aguapey VGA (Ituzaingó district) where grasslands were reduced to almost 50% of their original cover as a consequence of the expansion of tree plantations. This afforestation induced temperate grassland fragmentation, which initially included the perforation and subsequent attrition of grasslands. The structural changes within the Aguapey VGA were mainly a consequence of the national afforestation policy launched in the mid 1990s, which provided financial support for the development of the forestry industry; however, the traditional cattle ranching practices deeply rooted in the Aguapey region's culture may have partially inhibited the expansion of tree plantations within the area.

Although the ecological processes impacted by grassland afforestation are not directly studied, the clear evidence of grassland fragmentation within the Aguapey VGA described in this paper provides critical information to suggest that severe changes will have occurred in this ecosystem. These changes should be considered as early warning signs to develop conservation actions and protect this undervalued land cover.

Currently, slightly above three quarters of the remaining, most-well preserved, temperate grasslands of the Aguapey VGA are distributed towards the southern region within the Santo Tomé district (from 1083.83 km$^2$ of grassland cover recorded in 2020, 820.13 km$^2$ are distributed across Santo Tomé district; Tables 3 and 5). Since potentially irreparable changes have been shown to occur through afforestation of previous grassland areas over the space of a small number of years, and considering that it is highly probable that over the next years tree plantations will expand towards the southern region of the Aguapey VGA, urgent conservation land-use planning policies need to be developed to emphasise both the importance of the soil organic carbon stored in this region and its role as a habitat of globally endangered species. This is particularly prescient given the region's lack of formal protection and the new national policies promoting afforestation activity. These planning policies should promote the placement of forestry systems in areas that minimise their impact on existent temperate grassland ecosystems, their biodiversity, and the ecosystem services that they provide.

**Author Contributions:** Conceptualization, M.A. and N.G.B.; formal analysis, M.A.; investigation, M.A.; methodology, M.A. and N.G.B.; supervision, N.G.B. and M.B.; writing—original draft, M.A.; writing—review and editing, N.G.B. and M.B. All authors have read and agreed to the published version of the manuscript.

**Funding:** This research received no external funding.

**Institutional Review Board Statement:** Not applicable.

**Informed Consent Statement:** Not applicable.

**Data Availability Statement:** The data presented in this study are contained within the article and Appendices. Appendix A: Image Pre-Processing. Appendix B: Unsupervised classifications based on land covers' phenological signatures. Appendix C: Patch Level Landscape Metrics Analysis. Appendix D: Accuracy Assessment Analysis for Each Annual Period Based on Contingency Matrices. Any other data presented in this study and not included within the Appendices are available on request from the corresponding author.

**Acknowledgments:** We would like to thank Sebastián Cirignoli who provided key land cover information. We also thank three anonymous reviewers for their support and comments.

**Conflicts of Interest:** The authors declare no conflict of interest.

## Appendix A

*Image Pre-Processing*

Landsat Thematic Mapper (TM) and Landsat Operational Land Imager (OLI) Collection 1 Level 2 images were obtained from the public domain service of Earth Explorer USGS (https://earthexplorer.usgs.gov, accessed on 17 March 2020). The Landsat images selected (Path/Row 225/079) covered four annual periods: 1999–2000, 2006–2007, 2014–2015, and 2019–2020 (Table A1). All the selected scenes were cloud-free to minimize possible effects of the atmosphere on the image classification process [77].

No radiometric calibrations nor atmospheric corrections were applied given that the images were already pre-processed to bottom of the atmosphere (BOA) reflectance values (USGS, 2020). However, prior to analysis, all images were projected to the Universal Transverse Mercator System (UTM) zone 21S coordinate system, and subsequently, they were co-registered to sub-pixel accuracy to avoid geometric incongruity between images that may produce spurious classification results [78]. The image from June 1999 was taken as a reference; the remaining images were co-registered to the reference image using at least 30 ground control points spread throughout the scene. A second-order polynomial fit and the nearest neighbor method were applied in the rectification processes. In all cases, the root mean square error (RMSE) was less than 0.5 pixels. Images were pre-processed using ENVI software.

**Table A1.** Landsat images used for analysis.

| Annual Period | Satellite Imagery | Sensor | Acquisition Dates (Month/Year) |
|---|---|---|---|
| 1999–2000 | Landsat 5 | TM | 06/1999; 08/1999; 11/1999; 12/1999; 12/1999; 01/2000; 02/2000; 03/2000 |
| 2006–2007 | Landsat 5 | TM | 07/2006; 08/2006; 09/2006; 11/2006; 12/2006; 01/2007; 04/2007 |
| 2014–2015 | Landsat 8 | OLI | 06/2014; 08/2014; 10/2014; 11/2014; 12/2014; 03/2014; 04/2015; 05/2015 |
| 2019–2020 | Landsat 8 | OLI | 06/2019; 08/2019; 11/2019; 02/2020; 03/2020; 04/2020; 05/2020. |

## Appendix B

Landcovers' phenological signaturesUnsupervised classifications were performed to four NDVI temporal series (1999–2000; 2006–2007; 2014–2015; 2019–2020) following a phenological approach. Therefore, in order to identify the land covers classified in each period, phenological signatures were built for each one of the outcoming classes (Figure A1). Temperate grasslands showed a unimodal NDVI response, registering the lowest values in winter (from July to September), a steady increase until the peak in the

summer months (from January to March) and afterwards a slow NDVI decrease [37]; whereas tree plantations registered NDVI values above 0.7 throughout the year [13]. Other covers grouped all those covers that did not show a clear phenological pattern to be identified as either grasslands or tree plantations.

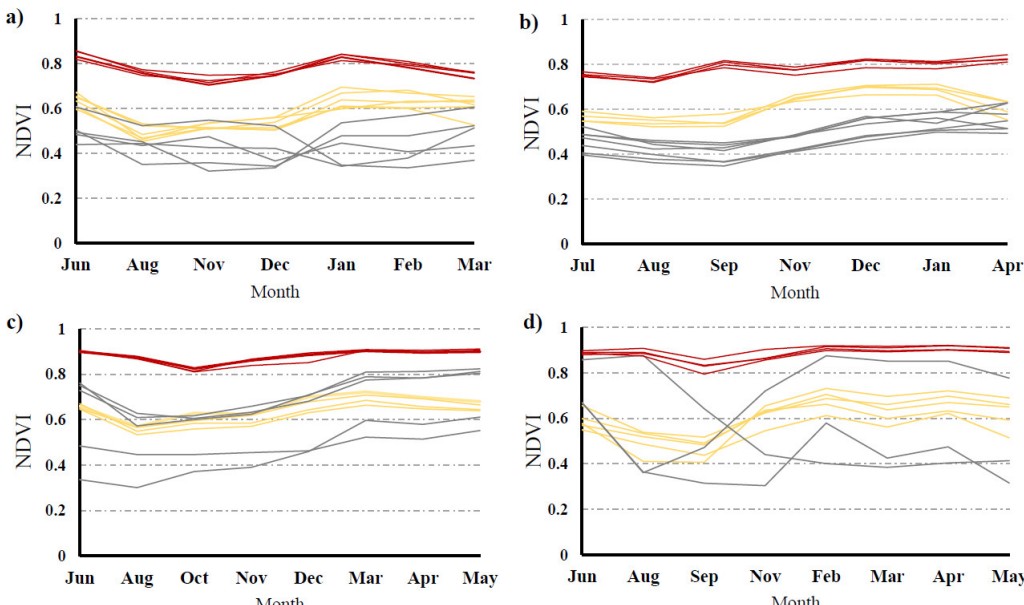

**Figure A1.** Example of phenological signatures of the outcoming classes obtained from ISODATA unsupervised classification for the Aguapey VGA in (**a**) 1999–2000, (**b**) 2006–2007, (**c**) 2014–2015 and (**d**) 2019–2020. Yellow lines represent grassland covers, red lines represent tree plantation covers (red lines), and grey lines represent other covers. Spectral signatures were randomly obtained from several regions of interest across the study area.

**Appendix C**

*Patch Level Landscape Metrics Analysis*

**Table A2.** Shapiro-Wilks's goodness-of-fit tests for the landscape metrics at patch level.

| Variable | n | Mean | St.Dv. | W | p |
|---|---|---|---|---|---|
| Area | 925 | 5.14 | 71.74 | 0.05 | <0.0001 |
| SHAPE | 925 | 1.37 | 0.82 | 0.37 | <0.0001 |
| ENN | 925 | 195.46 | 195.46 | 0.66 | <0.0001 |
| ECON | 925 | 48.33 | 40.5 | 0.83 | <0.0001 |

**Table A3.** Area's Kruskal Wallis tests between time periods.

| Annual Period | N | Mean | St. Dv. | H [1] | p | Ranks | | |
|---|---|---|---|---|---|---|---|---|
| 1999–2000 | 84 | 17.07 | 156.13 | 35.68 | <0.0001 | 408.95 | A [1] | |
| 2006–2007 | 418 | 2.68 | 48.21 | | | 469.27 | | B [1] |
| 2014–2015 | 225 | 4.99 | 60.13 | | | 493.2 | | B [1] |
| 2019–2020 | 198 | 5.47 | 69.02 | | | 543.94 | | B [1] |

[1] Means with a common letter are not significantly different (*p* > 0.05).

**Table A4.** Shape Index's Kruskal Wallis tests between time periods.

| Annual Period | N | Mean | St. Dv. | H | *p* | Ranks | | | |
|---|---|---|---|---|---|---|---|---|---|
| 1999–2000 | 84 | 1.31 | 1.17 | 20.25 | 0.0001 | 377.33 | A [1] | | |
| 2006–2007 | 418 | 1.37 | 1.20 | | | 446.08 | | B [1] | |
| 2014–2015 | 225 | 1.39 | 1.20 | | | 474.97 | | B [1] | C [1] |
| 2019–2020 | 198 | 1.45 | 1.26 | | | 521.46 | | | C [1] |

[1] Means with a common letter are not significantly different (*p* > 0.05).

**Table A5.** Euclidean Distance's Kruskal Wallis tests between time periods.

| Annual Period | N | Mean | St. Dv. | H | *p* | Ranks | | |
|---|---|---|---|---|---|---|---|---|
| 1999–2000 | 84 | 147.95 | 95.5 | 24.92 | <0.0001 | 414.54 | A [1] | |
| 2006–2007 | 418 | 161.68 | 133.99 | | | 425.99 | A [1] | |
| 2014–2015 | 225 | 235.28 | 223.83 | | | 524.05 | | B [1] |
| 2019–2020 | 198 | 231.92 | 271.79 | | | 490.31 | | B [1] |

[1] Means with a common letter are not significantly different (*p* > 0.05).

**Table A6.** Edge Contrast Index's Kruskal Wallis tests between time periods.

| Annual Period | N | Mean | St. Dv. | H | *p* | Ranks | | |
|---|---|---|---|---|---|---|---|---|
| 1999–2000 | 84 | 22.31 | 36.55 | 305.28 | <0.0001 | | A [1] | |
| 2006–2007 | 418 | 27.03 | 33.09 | | | | A [1] | |
| 2014–2015 | 225 | 73.14 | 30.48 | | | | | B [1] |
| 2019–2020 | 198 | 76.15 | 32.00 | | | | | B [1] |

[1] Means with a common letter are not significantly different (*p* > 0.05).

**Table A7.** Area's Kruskal Wallis tests between time periods for the Ituzaingó district.

| Annual Period | N | Mean | St. Dv. | H | *p* | Ranks | | | |
|---|---|---|---|---|---|---|---|---|---|
| 1999–2000 | 42 | 13.09 | 84.7 | 37.98 | <0.0001 | 304.44 | A [1] | | |
| 2006–2007 | 343 | 0.95 | 12.61 | | | 369.68 | | B [1] | |
| 2014–2015 | 186 | 1.57 | 11.32 | | | 391.43 | | | C [1] |
| 2019–2020 | 156 | 1.68 | 11.92 | | | 435.98 | | B [1] | C [1] |

[1] Means with a common letter are not significantly different (*p* > 0.05).

**Table A8.** Shape Index's Kruskal Wallis tests between time periods for the Ituzaingó district.

| Annual Period | N | Mean | St. Dv. | H | *p* | Ranks | | | |
|---|---|---|---|---|---|---|---|---|---|
| 1999–2000 | 42 | 1.35 | 1.37 | 17.31 | 0.0005 | 277.57 | A [1] | | |
| 2006–2007 | 343 | 1.38 | 0.76 | | | 346.04 | | B [1] | |
| 2014–2015 | 186 | 1.4 | 0.71 | | | 364.08 | | B [1] | |
| 2019–2020 | 156 | 1.47 | 0.88 | | | 410.61 | | | C [1] |

[1] Means with a common letter are not significantly different (*p* > 0.05).

**Table A9.** Euclidean Distance's Kruskal Wallis tests between time periods for the Ituzaingó district.

| Annual Period | N | Mean | St. Dv. | H | *p* | Ranks | | |
|---|---|---|---|---|---|---|---|---|
| 1999–2000 | 42 | 137.96 | 98.14 | 23.24 | <0.0001 | 294.54 | A [1] | |
| 2006–2007 | 343 | 159.22 | 125.69 | | | 329.03 | A [1] | |
| 2014–2015 | 186 | 224.19 | 208.43 | | | 394.13 | | B [1] |
| 2019–2020 | 156 | 245.62 | 287.98 | | | 410.11 | | B [1] |

[1] Means with a common letter are not significantly different (*p* > 0.05).

**Table A10.** Edge Contrast Index's Kruskal Wallis tests between time periods for the Ituzaingó district.

| Annual Period | N | Mean | St. Dv. | H | p | Ranks | | | |
|---|---|---|---|---|---|---|---|---|---|
| 1999–2000 | 42 | 29.37 | 39.99 | 264.26 | <0.0001 | 234.77 | A [1] | | |
| 2006–2007 | 343 | 25.64 | 31.43 | | | 255.02 | A [1] | | |
| 2014–2015 | 186 | 73.73 | 29.23 | | | 479.85 | | B [1] | |
| 2019–2020 | 156 | 77.63 | 29.84 | | | 500.09 | | B [1] | |

[1] Means with a common letter are not significantly different ($p > 0.05$).

**Table A11.** Area's Kruskal Wallis tests between time periods for the Santo Tomé district.

| Annual Period | N | Mean | St. Dv. | H | p |
|---|---|---|---|---|---|
| 1999–2000 | 44 | 19.98 | 132.08 | 2.23 | 0.5233 |
| 2006–2007 | 75 | 12.69 | 81.11 | | |
| 2014–2015 | 39 | 21.27 | 149.78 | | |
| 2019–2020 | 46 | 17.83 | 117.62 | | |

**Table A12.** Shape Index's Kruskal Wallis tests between time periods for the Santo Tomé district.

| Annual Period | N | Mean | St. Dv. | H | p |
|---|---|---|---|---|---|
| 1999–2000 | 44 | 1.36 | 1.04 | 4.35 | 0.2136 |
| 2006–2007 | 75 | 1.36 | 1.04 | | |
| 2014–2015 | 39 | 1.32 | 0.91 | | |
| 2019–2020 | 46 | 1.37 | 1.01 | | |

**Table A13.** Euclidean Distance's Kruskal Wallis tests between time periods for the Santo Tomé district.

| Annual Period | N | Mean | St. Dv. | H | p |
|---|---|---|---|---|---|
| 1999–2000 | 44 | 154.58 | 92.16 | 4.98 | 0.1693 |
| 2006–2007 | 75 | 164.82 | 157.82 | | |
| 2014–2015 | 39 | 194.65 | 144.21 | | |
| 2019-2020 | 46 | 179.19 | 193.29 | | |

**Table A14.** Edge Contrast Index's Kruskal Wallis tests between time periods for the Santo Tomé district.

| Annual Period | N | Mean | St. Dv. | H | p | Ranks | | | |
|---|---|---|---|---|---|---|---|---|---|
| 1999–2000 | 44 | 15.07 | 31.1 | 47.65 | <0.0001 | 67.28 | a [1] | | |
| 2006–2007 | 75 | 33.72 | 37.4 | | | 96.33 | | b [1] | |
| 2014–2015 | 39 | 68.67 | 35.94 | | | 143.48 | | | c [1] |
| 2019–2020 | 46 | 68.09 | 38.43 | | | 143.98 | | | c [1] |

[1] Means with a common letter are not significantly different ($p > 0.05$).

## Appendix D

*Accuracy Assessment Analysis for Each Annual Period Based on Contingency Matrices*

**Table A15.** Error matrix for the unsupervised classification for the period 1999–2000 based on 200 random points distributed across the Aguapey VGA. The diagonal contains correctly classified pixels.

| | Ground Truth Data | | | | |
|---|---|---|---|---|---|
| **Classified Data** | **Grasslands** | **Tree Plantations** | **Other Covers** | **Total** | **User Accuracy (%)** |
| Grasslands | 157 | 0 | 12 | 169 | 92.9 |
| Tree plantations | 0 | 4 | 0 | 3 | 100 |
| Other covers | 4 | 1 | 22 | 27 | 81.48 |
| Total | 161 | 5 | 34 | 200 | |
| Producer accuracy (%) | 97.51 | 80 | 64.71 | | |
| Overall accuracy | 0.92 | | | | |

**Table A16.** Error matrix for the unsupervised classification for the period 2006–2007 based on 200 random points distributed across the Aguapey VGA. The diagonal contains correctly classified pixels.

| | Ground Truth Data | | | | |
|---|---|---|---|---|---|
| **Classified Data** | **Grasslands** | **Tree Plantations** | **Other Covers** | **Total** | **User Accuracy (%)** |
| Grasslands | 129 | 0 | 10 | 139 | 92.81 |
| Tree plantations | 0 | 10 | 0 | 10 | 100 |
| Other covers | 6 | 2 | 43 | 51 | 84.13 |
| Total | 135 | 12 | 53 | 200 | |
| Producer accuracy (%) | 95.56 | 83.33 | 81.13 | | |
| Overall accuracy | 0.91 | | | | |

**Table A17.** Error matrix for the unsupervised classification for the period 2014–2015 based on 1000 random points distributed across the Aguapey VGA. The diagonal contains correctly classified pixels.

| | Ground Truth Data | | | | |
|---|---|---|---|---|---|
| **Classified Data** | **Grasslands** | **Tree Plantations** | **Other Covers** | **Total** | **User Accuracy (%)** |
| Grasslands | 655 | 3 | 37 | 695 | 94.24 |
| Tree plantations | 39 | 223 | 8 | 270 | 82.59 |
| Other covers | 27 | 0 | 8 | 35 | 22.86 |
| Total | 721 | 226 | 53 | 1000 | |
| Producer accuracy (%) | 90.85 | 98.67 | 15.09 | | |
| Overall accuracy | 0.89 | | | | |

**Table A18.** Error matrix for the unsupervised classification for the period 2019–2020 based on 200 random points distributed across the Aguapey VGA. The diagonal contains correctly classified pixels.

| | Ground Truth Data | | | | |
|---|---|---|---|---|---|
| **Classified Data** | **Grasslands** | **Tree Plantations** | **Other Covers** | **Total** | **User Accuracy (%)** |
| Grasslands | 126 | 2 | 16 | 144 | 87.5 |
| Tree plantations | 0 | 51 | 1 | 52 | 98.08 |
| Other covers | 2 | 0 | 2 | 4 | 50 |
| Total | 128 | 53 | 19 | 200 | |
| Producer accuracy (%) | 96.22 | 98.43 | 10.53 | | |
| Overall accuracy | 0.9 | | | | |

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
