# Peer review of "Temperate Grassland Afforestation Dynamics in the Aguapey Valuable Grassland Area between 1999 and 2020: Identifying the Need for Protection"

_remotesensing, doi:10.3390/rs14010074_

Round 1

Reviewer 1 Report

The article focused on the landscape change of temperate grasslands, and provided a vivid picture of historical trends. Results and discussions are complete. Here are some suggestions for your reference:

  1. Introduction. Natural system is under human disturbance. Temperate grasslands are a part of nature. What are the main threats, and negative outcomes of grassland change in general? It would be clearer if it is introduced at the beginning of introduction.
  2. Figure 3. Although the transition probability suggests the conversion of land covers from the perspective of proportion, differences among area of land covers are significant. The analysis of transition including the area of land covers would be better. For example, from 1999 to 2006, 57% of other covers converted to grassland (59.24km2), and 22% of grassland converted to other covers (315.66km2). The loss of grassland is much more than that of other covers.
  3. Page 12, 3.4 District scale analysis. The heterogeneity is mainly represented by the area change. Are there landscape (PD/PS…) differences between the two districts?
  4. Line 576-585. There is no doubt that grasslands represent large carbon sinks. However, does it superior to forest in the aspect of carbon storage? What's the advantage of grasslands comparing to forest?
  5. Details: Line 262, formula (2), meaning of Pij needs to clarify; Figure 2, demonstrating the change of tree plantation among periods might be clearer.

Reviewer 2 Report

This is a fine piece of research in which the authors use of rather simple image analysis techniques in an elegant manner to provide relevant results concerning the fragmentation and attrition of temperate grasslands in a region in Argentina. Overall, the paper is well written and the contents are well presented. I have no objections or comments regarding the selection of methods, materials, or the interpretation of the results and, consequently, I believe the manuscript should be accepted for publication in Remote Sensing.

Author Response

The authors really appreciate the reviewer's comments.

Reviewer 3 Report

The only thing I want to see a clarification is the reason to have the 7x7 pixels smooth. What is the reason for 7x7?
